# Evaluation of *(R)*-[^11^C]PK11195 PET/MRI for Spinal Cord-Related Neuropathic Pain in Patients with Cervical Spinal Disorders

**DOI:** 10.3390/jcm12010116

**Published:** 2022-12-23

**Authors:** Makoto Kitade, Hideaki Nakajima, Tetsuya Tsujikawa, Sakon Noriki, Tetsuya Mori, Yasushi Kiyono, Hidehiko Okazawa, Akihiko Matsumine

**Affiliations:** 1Department of Orthopaedics and Rehabilitation Medicine, Faculty of Medical Sciences, University of Fukui, Fukui 910-1193, Japan; 2Department of Radiology, Faculty of Medical Sciences, University of Fukui, Fukui 910-1193, Japan; 3Faculty of Nursing and Social Welfare Science, Fukui Prefectural University, Fukui 910-1195, Japan; 4Biomedical Imaging Research Center, Faculty of Medical Sciences, University of Fukui, Fukui 910-1193, Japan

**Keywords:** [^11^C]PK11195 PET/MRI, spinal cord, neuropathic pain, activated microglia, translocator protein 18 kDa (TSPO), spinal disorders

## Abstract

Activated microglia are involved in secondary injury after acute spinal cord injury (SCI) and in development of spinal cord-related neuropathic pain (NeP). The aim of the study was to assess expression of translocator protein 18 kDa (TSPO) as an indicator of microglial activation and to investigate visualization of the dynamics of activated microglia in the injured spinal cord using PET imaging with *(R)*-[^11^C]PK11195, a specific ligand for TSPO. In SCI chimeric animal models, TSPO was expressed mainly in activated microglia. Accumulation of *(R)*-[^3^H]PK11195 was confirmed in autoradiography and its dynamics in the injured spinal cord were visualized by *(R)*-[^11^C]PK11195 PET imaging in the acute phase after SCI. In clinical application of *(R)*-[^11^C]PK11195 PET/MRI of the cervical spinal cord in patients with NeP related to cervical disorders, uptake was found in cases up to 10 months after injury or surgery. No uptake could be visualized in the injured spinal cord in patients with chronic NeP at more than 1 year after injury or surgery, regardless of the degree of NeP. However, a positive correlation was found between standardized uptake value ratio and the severity of NeP, suggesting the potential of clinical application for objective evaluation of chronic NeP.

## 1. Introduction

Spinal cord injury (SCI) and degenerative compressive myelopathy (DCM) result in loss of motor function and normal sensation, and often cause debilitating neuropathic pain (NeP), including allodynia. At-level pain, which presents within two segments above and below the level of injury, occurs in 37–50% of patients with SCI, whereas below-level pain, which presents at least three segments below the neurological level of injury, occurs in 76–83% of patients [1,2,3]. In patients with DCM, 41% with cervical spondylotic myelopathy and 60% with ossification of the longitudinal ligament have postoperative residual NeP [4]. These symptoms are well known to be associated with significant impairment of health-related quality of life and increased economic costs [5,6,7,8,9]. A nationwide survey of spinal cord-related pain in Japan found that 43.0% of patients presented with allodynia and about 25% of the patients with allodynia had impaired employment and activities of daily living [10,11]. Therefore, there is a need to establish objective methods to evaluate subjective pain, in order to understand the pathophysiology and determine the efficacy of treatment.

Infiltration of neutrophils and macrophages, together with resident microglia, are associated with inflammatory responses after SCI. Activation of glial cells reduces axonal regrowth, which may ultimately result in exacerbation of neurological dysfunction due to cavitation and cyst formation at the lesion center [12]. In addition, functional abnormalities of neurons in the spinal cord and brain are caused by signals from glial cells, and a persistent activated state of these cells plays an important role in chronic NeP, as well as having both neuroprotective and neurodamaging effects [13,14]. This active chronic inflammatory state caused by the neuron-glial interaction is referred to as neuroinflammation and is involved in the pathogenesis of spinal cord-related NeP, making it a therapeutic target for NeP.

Recent developments of neuroimaging techniques including positron emission tomography (PET)/magnetic resonance imaging (MRI) have facilitated in vivo imaging of biological pathways in disease. To visualize the dynamics of glial activation, a key stage in spinal cord regeneration and NeP, we examined *(R)*-[^11^C]PK11195 (1-[2-chlorophenyl]-N-methyl-N-[1-methylpropyl]-3-isoquinoline carboxamide) as a potential indicator of glial cell activation. PK11195 is a specific ligand for translocator protein 18 kDa (TSPO, formerly known as peripheral benzodiazepine receptor) [15,16]. TSPO is located in the mitochondria of glial cells in the central nervous system (CNS) and is expressed in the outer membrane of the mitochondria, reflecting glial activation, after nerve damage or degeneration [17,18].

Based on the characteristics of PK11195, *(R)*-[^11^C]PK11195 PET has been used to assess the degree of neuroinflammation in neurodegenerative diseases [19], including Alzheimer’s disease [20], multiple sclerosis [21], and complex regional pain syndrome [22]. However, *(R)*-[^11^C]PK11195 PET imaging of the spinal cord in patients with NeP has not been reported. Therefore, the aims of this study were to evaluate visualization of glial activation by *(R)*-[^11^C]PK11195 PET in a SCI model, and to examine the clinical significance of *(R)*-[^11^C]PK11195 PET/MRI for patients with NeP due to cervical SCI and DCM.

## 2. Materials and Methods

### 2.1. Experimental Animals 

The study was conducted in male Sprague Dawley rats (Nihon SLC, Shizuoka, Japan) (*n* = 45; age >18 weeks, mean body weight 627.5 g (504–795) for PET imaging; age 10–12 weeks, mean body weight 374.8 g (361–406) for immunohistochemistry and autoradiography) (Table 1), and male rats with enhanced expression of green fluorescent protein driven by the cytomegalovirus early enhancer β-actin (CAG) transgene (CAG-EGFP rat; Nihon SLC, Shizuoka, Japan) (*n* = 15; age 8–10 weeks, mean body weight 352.3 g). 

### 2.2. Contusion Spinal Cord Injury Model

Under deep anesthesia with 2% isoflurane (Forane; Abbot Japan, Osaka, Japan), the rat spinal cord was exposed by laminectomy at the C4 vertebral level. An Infinite Horizon Impactor (Precision Systems and Instrumentation LLC, Fairfax Station, VA, USA) was used with an impact force of 200 kilodynes to produce a contusion SCI. Rats in the sham SCI group underwent laminectomy only at C4, with no SCI. 

### 2.3. Bone Marrow-Chimeric Rats

Bone marrow-chimeric rats were made using highly purified, genetically marked bone marrow cells [14,23,24], which allowed studies to be limited to hematopoietic lineages. Unfractionated marrow cells (5.0 × 10^6^ cells) isolated from a donor CAG-EGFP transgenic rat were injected intravenously into the tail vein of a recipient rat that had been irradiated at 10.0 Gy for 45 min with the head protected with a lead plate. After 4 weeks, dual-laser fluorescence-activated cell sorting (FACS Calibur, BD Biosciences, San Jose, CA, USA) was performed using peripheral blood from recipient rats to evaluate engraftment and induction of chimerism by identification of donor GFP-positive bone marrow-derived cells. Bone marrow-chimeric rats were used in SCI surgery at age 10–12 weeks.

### 2.4. Immunohistochemistry

Under anesthesia, rats were perfused intracardially with ice-cold phosphate-buffered saline (PBS, followed by fixation with 4% paraformaldehyde in 0.1 M PBS at days 1, 4, 7, 14 and 28 post-SCI (*n* = 3 at days 1, 7, 14 and 28; *n* = 7 at day 4). After a few hours in the fixative solution, tissue samples were left to stand in 10% sucrose/0.1 M PBS at 4 °C for 24 h, and then in 20% sucrose/0.1 M PBS for another 24 h. Perfused segments of the spinal cord (C1–T2) from separate animals were embedded in optimal cutting temperature compound (Sakura Finetek, Torrance, CA, USA) and cut on a cryostat for preparation of axial sections for post-SCI assessment of the dorsal horn and gray matter. The 10-μm frozen sections were serially mounted on glass slides and immersed in 0.3% Triton X-100 in PBS for 4 min on ice. The following primary antibodies were diluted in Antibody Diluent with Background Reducing Components (Dako Cytomation, Glostrup, Denmark) and applied overnight at 4 °C: mouse anti-CD11b (1:200, Abcam, Cambridge, UK); rabbit anti-TSPO (PBR; 1:200, ab154878, Abcam); mouse anti-GFAP (1:200, ab10062, Abcam); rabbit anti-Neu N (1:200, Abcam); and mouse anti-CC1 (1:200, Abcam). Sections were then incubated for 1 h at room temperature with Alexa fluor-conjugated 488 or 568 secondary antibodies (1:500; Abcam), followed by rinsing three times in PBS. Immunopositivity was examined on images collected using a fluorescence microscope (Olympus AX80; Olympus Optical). Positive cells were counted in 10 randomly chosen transverse sections within 5 mm rostral and caudal to the lesion, with capture of 20 non-overlapping high-magnification (×200) photomicrographs.

### 2.5. Ex Vivo (R)-[^3^H]PK11195 Autoradiography

In autoradiography, rats were intravenously injected with *(R)*-[^3^H]PK11195 (18 MBq, 85.5 Ci/mmol, 2993 GBq/mmol; Perkin-Elmer, St. Louis, MO, USA) into the tail vein. Rats were sacrificed 1 h after injection under deep anesthesia with 2% isoflurane on days 0 (naive), 4, 14 and 28, followed by fixation with 4% paraformaldehyde (*n* = 3 for each time point). The C1-L2 segments of the spinal cord were rapidly removed, formalin-fixed and paraffin-embedded. Transverse sections of 10 μm were prepared using a Microtome (REM-710, SBF240W) and mounted on glass slides. A photographic emulsion (Sakura NR-H2) was applied to the slides and the mixture was left in a dark room for one month. The anatomy of the section was confirmed by staining with Hematoxylin. The number of *(R)*-[^3^H]PK11195-positive cells in the spinal dorsal horn was counted in 10 randomly chosen transverse sections up to 5 mm rostral and caudal to the lesion, using 20 high-magnification (×200) photomicrographs.

### 2.6. (R)-[^11^C]PK11195 PET Imaging in the Rat Spinal Cord Injury Model

[^11^C]PK11195 was synthesized from *(R)*-*N*-desmethyl-PK1195 (1 mg, ABX, Germany) with a [^11^C]methyl iodide. Radiochemical purity of [^11^C] PK11195 was >99% and specific activity was 33–99 GBq/µmoL. Rats were intravenously injected with *(R)*-[^11^C]PK11195 (18.5 MBq) into the tail vein at 4 and 14 days after SCI and *(R)*-[^11^C]PK11195 PET (SHR-41000, Hamamatsu Photonics) was performed with the head and extremities fixed to a special pedestal for imaging. The spatial resolution at the center of field of view (FOV) was <2.0 mm full width at half maximum (FWHM). Data were dynamically acquired for 60 min (*n* = 5 at day 4, *n* = 3 at day 14, *n* = 3 for normal rats without surgery, *n* = 3 for sham-operated rats). Separately, multi-slice CT scans (Hitachi Medical) were recorded, and PET and CT fusion images were created manually using image analysis software (PMOD3.6). Based on the fusion image, a region of interest (ROI) with a lateral diameter of 3 mm at the C4 lamina level was set on the damaged spinal canal. The standardized uptake value (SUV) was calculated as a semi-quantitative value using the following equation: SUV = (tissue activity (kBq/mL) × body weight (kg)) / injected *(R)*-[^11^C]PK11195 dose (MBq). SUV_max_, the maximal ROI count, was used as the tissue activity.

### 2.7. (R)-[^11^C]PK11195 PET/3T-MRI in Patients with Spinal Disorder-Related Neuropathic Pain

*(R)*-[^11^C]PK11195 PET/3T-MRI was performed for a healthy subject and 7 patients with NeP related to spinal disorders including SCI, cervical spondylotic myelopathy (CSM), and ossification of the posterior longitudinal ligament (OPLL). Injection of *(R)*-[^11^C]PK11195 (400–800 MBq) was performed over 5 s into the antecubital vein. At 50 min post-injection, static scans were collected over 10 min using a GE Signa PET-MR scanner (GE Healthcare, Milwaukee, WI, USA). The scanner permits PET data acquisition of 89 image slices in a three-dimensional acquisition mode with interslice spacing of 2.78 mm [25]. Performance tests showed that the intrinsic resolution of PET images ranged from 4.2–4.3 mm FWHM in the transaxial direction. The PET/MRI scanner was calibrated with a dose-calibrator (CRC-12, Capintec Inc., NJ, USA) using a pool phantom and ^18^F-solution prior to the *(R)*-[^11^C]PK11195 PET/MRI study, according to the scanner manufacturer’s guidelines [25,26]. Multiple MRI sequences were acquired during PET scan simultaneously, including T1WI, T2WI (sagittal, axial), and DIXON for PET attenuation correction (AC) image data. PET images were reconstructed from list mode PET and MR-AC data using the 3D ordered subset expectation maximization (OSEM) method with the following parameter: subset, 32; iteration, 3; transaxial post-Gaussian filter cutoff, 4 mm in 256 mm FOV and 1 × 1 mm^2^ pixel size. The PET image was converted to the SUV image using the injection dose and body weight of each subject and decay of the tracer was corrected to the injection time. 

Quantitative analysis was performed using round ROIs of diameter 4 mm on the spinal cord. Using these ROIs, the SUV ratio (SUVR) was calculated as SUV_max_ for cervical spinal cord lesions with increased MRI signal intensity in the sagittal plane/SUV_mean_ for the normal cervical spinal cord at the C7/T1 level [27]. The Neuropathic Pain System Inventory (NPSI) was used at the time of *(R)*-[^11^C]PK11195 PET/MRI to assess the degree of NeP [28,29]. Each patient gave written informed consent prior to the study. The protocol was approved by the Human Ethics Review Committee of Fukui University Medical Faculty (Approval No. 20160053) and followed the Clinical Research Guidelines of the Ministry of Health, Labor, and Welfare of Japan.

### 2.8. Statistical Analysis

All values are expressed as mean ± SD. Intergroup differences were evaluated by Wilcoxon signed rank test and Mann–Whitney U-test, with *p* < 0.05 denoting significance. Pearson correlation analysis was used to examine the relationship among time after injury/surgery, NPSI scores, and SUVR. All statistical tests were performed using SPSS software version 24.0 (SPSS, Chicago, IL, USA).

## 3. Results

### 3.1. Cell Source of TSPO in the Rat Spinal Cord Injury Model

To identify cells expressing TSPO, immunohistochemical analysis was performed for TSPO and several cell markers. In double immunostaining, most TSPO-positive cells merged with CD11b, a marker of activated microglia and/or macrophages, and few merged with other cell markers (NeuN for neurons, CC1 for oligodendrocytes and GFAP for astrocytes) (Figure 1). To distinguish hematogenous macrophages from activated microglia expressing TSPO, bone marrow-chimeric rats were prepared with GFP-labeled hematogenous cells. The number of TSPO-positive cells at the injured site peaked at 4 days after SCI and significantly fewer TSPO-positive cells merged with GFP over time (Figure 2). These results suggest that TSPO was mainly expressed in activated microglia.

### 3.2. (R)-[^11^C]PK11195 PET in the Rat Spinal Cord Injury Model

The mean SUV_max_ at the C4 level was 1.22 ± 0.27 in the normal group, 1.38 ± 0.23 in the sham group, 2.89 ± 0.53 in the 4-day post-SCI group, and 1.88 ± 0.41 in the 14-day post-SCI group. There was a significant increase in *(R)*-[^11^C]PK11195 uptake at the lesion level in the SCI groups compared to those in the control and sham groups (Figure 3).

### 3.3. Accumulation of (R)-[^3^H]PK11195 after Spinal Cord Injury

*(R)*-[^3^H]PK11195 autoradiography was performed to assess accumulation of *(R)*-[^3^H]PK11195 in the injured spinal cord (Figure 4). Positive cells with black dots were observed in the dorsal horn of the injured spinal cord. The number of *(R)*-[^3^H]PK11195-positive cells peaked at 4 days after SCI and gradually decreased thereafter. There was no *(R)*-[^3^H]PK11195 accumulation in control rats.

### 3.4. (R)-[^11^C]PK11195 PET/3T-MRI in Patients with Spinal Disorder-Related Neuropathic Pain

Demographic data for the patients are shown in Table 2 and examples of images are shown in Figure 5. A healthy subject showed no *(R)*-[^11^C]PK11195 uptake (Figure 5A). SUVR was 1.19 to 2.05 in patients with lesions with higher signal intensity in the sagittal plane on T2-weighted MRI. Visualized uptake was found in cases up to 10 months after injury or surgery (Figure 5B,C). There was no significant *(R)*-[^11^C]PK11195 uptake in a patient at 9 years after SCI, despite the presence of severe NeP (Case 6: NPSI score 30.5) (Figure 5D). Similarly, in patients examined 1 to 4 years after injury or surgery (Cases 4–6), there was no *(R)*-[^11^C]PK11195 visualized uptake regardless of the degree of NeP. However, there was a negative correlation (R = 0.78) between SUVR and time after injury/surgery (Figure 6A). Furthermore, there was a positive correlation (R = 0.91) between SUVR and NPSI, with the exception of case 7 (SCI 9 years previously induced spinal cord atrophy) (Figure 6B).

## 4. Discussion

The aim of the current study was to assess TSPO expression as an indicator of microglial activation and to investigate visualization of the dynamics of activated microglia using PET imaging with *(R)*-[^11^C]PK11195, a specific ligand for TSPO. The results suggested that TSPO is expressed mainly in activated microglia and that its dynamics in the injured spinal cord can be visualized by PET imaging in the acute phase after SCI using *(R)*-[^11^C]PK11195. A trial of *(R)*-[^11^C]PK11195 PET/3T-MRI of the cervical spinal cord in patients with NeP related to cervical disorders showed limited visualization in the chronic phase. However, a positive correlation was found between SUVR and severity of NeP, except in case of long-term spinal cord atrophy after SCI, suggesting the potential of clinical application for objective evaluation of chronic NeP.

TSPO is of particular importance in preserving normal physiological functions, such as initiation of programmed cell death and regulation of gene expression in response to oxidative stress in the CNS [30,31,32,33]. TSPO is also a marker for glial activation, especially activated microglia, but it has been suggested that TSPO is expressed on multiple different immune cells, including reactive astrocytes [17,34]. A significant increase of TSPO in astrocytes has been found in neurodegenerative diseases, such as Alzheimer’s disease and amyotrophic lateral sclerosis [35].

Given the differences in the roles of microglia and astrocytes in NeP, it is important to assess which types of glial cells contribute to TSPO expression in the injured spinal cord. In addition, distinction between activated microglia and hematogenous macrophages is also important in terms of assessment of NeP. A study in chimeric spinal hyperostotic mice (*ttw*/*ttw*) suggested correlations of chronic NeP in long-term compression of the spinal cord with infiltration of macrophages, activation of microglial cells, and resulting blood-spinal cord barrier damage, together with overexpression of pain-related molecules in these cells [36]. There are no specific antibodies that distinguish hematogenous macrophages from microglia [37]. Therefore, in this study, a GFP-positive bone marrow-chimeric rat was generated to assess the source of TSPO expression in an injured spinal cord. TSPO was detected mainly in activated microglia and some hematogenous macrophages, but not in neurons, oligodendrocytes, and astrocytes. 

Overexpression of TSPO is linked to several pathological conditions, and this makes the protein a particularly useful biomarker for diagnosis, prediction of outcomes, and evaluation of treatment efficacy [38]. PK11195 is a high affinity ligand for TSPO, and in the current study uptake was detected in the injured spinal cord at 4 and 14 days after SCI using *(R)*-[^11^C]PK11195 PET imaging. In addition, accumulation of *(R)*-[^11^C]PK11195 in the dorsal horn of the injured spinal cord was confirmed by spinal *(R)*-[^3^H]PK11195 autoradiography. An association between activation of dorsal horn microglia and tactile allodynia has been found, especially in the chronic phase [39], and glial activation can be quantified using imaging of the lumbar spinal cord of rats with partial sciatic nerve ligation on days 7 and 14 after injury [40]. In contrast, a recent study in the same rat model found that in vivo *(R)*-[^11^C]PK11195 and *(R)*-[^18^F]F-DPA (N,N-diethyl-2-(2-(4-([^18^F]fluoro)phenyl)- 5,7-dimethylpyrazolo[1,5-a] pyrimidin-3-yl) acetamide) PET imaging failed to show tracer accumulation due to background noise, along with spillover from the vertebral body, although ex vivo autoradiography showed higher lesion-to-background uptake with both tracers [41]. Strong uptake of TSPO PET tracers in bone marrow (part of the vertebral body) has been associated with hematopoietic stem cells in the medulla [42]. In the current study, background noise was less prominent in the rat model, but *(R)*-[^11^C]PK11195 PET/MRI in patients with NeP had a higher SUV in the vertebral body. Scatter may underlie this phenomenon, and this is likely to be more important in high resolution images [43]. In our study, background noise from the vertebral bodies had little affect the assessment of *(R)*-[^11^C]PK11195 uptake in the cervical spinal cord.

In the clinical trial of *(R)*-[^11^C]PK11195 PET/3T-MRI for patients with NeP, uptake was found in cases up to 10 months after injury or surgery, but not in cases at more than 1 year after these events. Although SUVR was decreasing over time after injury or surgery, a positive correlation was found between SUVR and NPSI, suggesting the potential of clinical application for objective assessment of NeP. Use of *(R)*-[^11^C]PK11195 is widespread, as the prototypical first-generation PET radioligand, but low sensitivity and poor quantification limit its utility. To overcome these limitations, several second-generation TSPO radioligands have been developed, including *(R)*-[^18^F]FEPPA [44], *(R)*-[^11^C]PBR06 [45], *(R)*-[^18^F]PBR111 [46], *(R)*-[^11^C]AC5216 [47] and *(R)*-[^18^F]PBR28 [48]. These ligands have higher affinity and selectivity, and improved physicochemical and pharmacokinetic properties, but are also sensitive to SNP rs6971 in the TSPO gene. Thus, binding affinity is related to the presence of SNP rs6921, and a similar TSPO density with a different genotype gives a different PET signal [46]. Development of third-generation TSPO radioligands such as *(R)*-[^18^F]GE180 [49] and *(R)*-[^11^C]ER176 [50] had the goals of high TSPO binding selectivity and low SNP sensitivity. However, there is still no ideal TSPO radioligand and further work is required to improve the utility of TSPO imaging in clinical settings [38].

The limitation of visualization of *(R)*-[^11^C]PK11195 for NeP in the current study may be due less to vertebral background noise making it difficult to assess spinal cord uptake, and more to the limitations of *(R)*-[^11^C]PK11195 in capturing spinal cord activated microglia in the low inflammatory state of the chronic phase. It is also possible that activated microglia in the injured spinal cord are less involved in the pathogenesis of chronic NeP. Spinal cord microglia activation is well understood to have a key role in nerve injury-induced central pain sensitization [51], and the distribution of activated microglia at the lumbar enlargement after SCI is important in the pathomechanism of below-level NeP [13,23]. These studies indicate that microglia activation in the dorsal horn and lumbar enlargement peaks in the acute and subacute phases after nerve injury. Microglial activation in specific brain sub-regions (medial prefrontal cortex, amygdala, and hippocampus) in the chronic phase after nerve injury has also been found in more recent studies [52,53]. These studies suggest that nerve injury-induced mechanisms of microglia activation in the brain differ from those in spinal cord microglia, and this warrants further study. In a rat model of temporal lobe epilepsy, early and chronic TSPO expression showed no correlation, suggesting that factors that induce microglial activation may differ depending on the disease phase [54]. 

This study has certain limitations, including the lack of evaluation in an animal model in the chronic phase after SCI, the small number of patients in the clinical study, and the use of a first-generation TSPO radioligand. Thus, further investigations using PET/MRI neuroimaging in brain sub-regions, including assessment of ideal TSPO radioligands, are needed to understand the roles and mechanisms of microglia in the development of chronic NeP and to assess the potential for visualization of chronic NeP. Despite these limitations, the results of the study provide important insights into the association between microglial activation in the injured spinal cord and chronic NeP, and may guide future directions for establishment of objective assessments of NeP through visualization of spinal glial cell activation.

## 5. Conclusions

In a rat SCI model, expression of TSPO was detected mainly in activated microglia, and *(R)*-[^11^C]PK11195 PET imaging was able to visualize the dynamics of activated microglia in the acute and subacute phases. In clinical application of *(R)*-[^11^C]PK11195 PET/MRI, visualized uptake was only detected in cases up to 10 months after injury or surgery, with no uptake in the injured spinal cord for patients with chronic NeP at over 1 year, regardless of the degree of NeP. However, there was a positive correlation between SUVR and NPSI, except in case of long-term spinal cord atrophy after injury, suggesting the potential of clinical application for objective evaluation of chronic NeP. The cause of this negative result for visualization of chronic NeP is unclear. Besides the problems associated with TSPO radioligands, it raises the possibility that activated microglia at the site of injury may play a lesser role in chronic NeP. Further studies of NeP visualization using PET/MRI are needed, with use of different TSPO radioligands and evaluation of brain sub-regions, in addition to the spinal cord and lumbar enlargement.

## Figures and Tables

**Figure 1 jcm-12-00116-f001:**
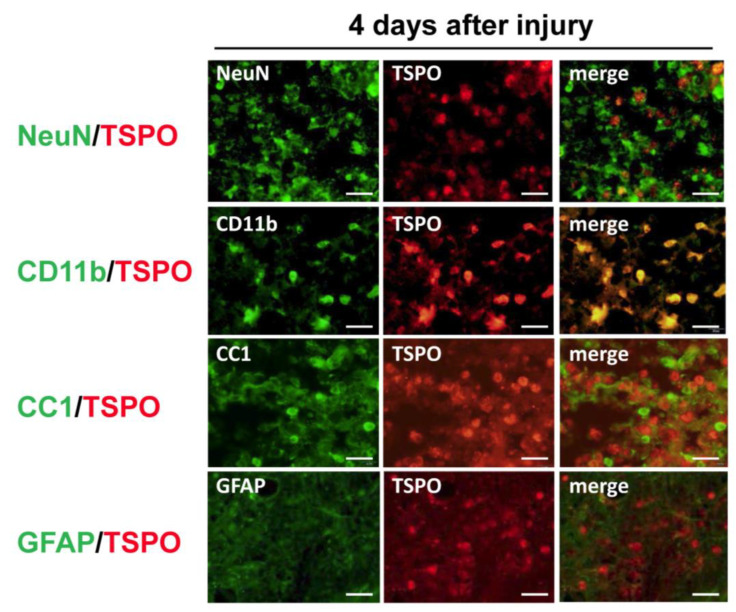
Colocalization of cell-specific markers (NeuN for neurons, CD11b for microglia/macrophages, CC1 for oligodendrocytes and GFAP for astrocytes) and translocator protein 18 kDa (TSPO) in the injured spinal cord at 4 days after spinal cord injury. Most TSPO merged with CD11b, a marker of activated microglia and/or macrophages. TSPO did not colocalize with markers for neurons, oligodendrocytes, and astrocytes. Scale bars = 20 μm.

**Figure 2 jcm-12-00116-f002:**
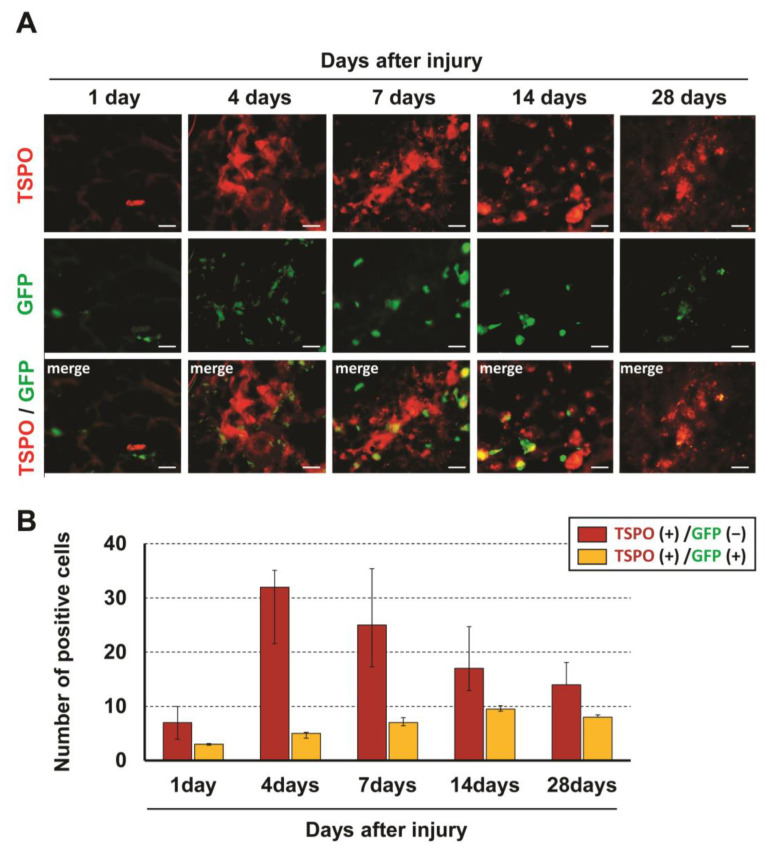
Immunofluorescent staining showing GFP-positive/TSPO-positive and GFP-negative/TSPO-positive cells at the injured site at 1, 4, 7, 14 and 28 days after SCI (**A**). The number of TSPO-positive cells at the injured site peaked at 4 days after SCI and significantly few-er TSPO-positive cells merged with GFP over time (**B**). Scale bars = 20 μm.

**Figure 3 jcm-12-00116-f003:**
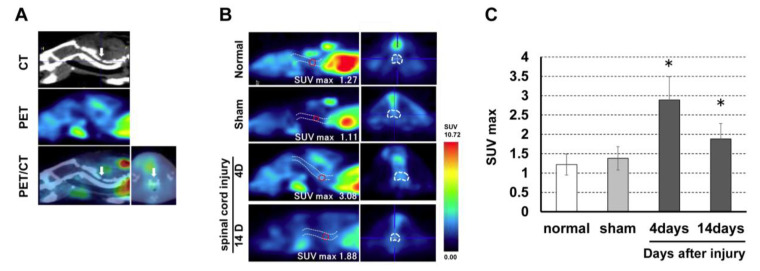
(**A**) PET and CT fusion images were created to detect the laminectomy level. (**B**) *(R)*-[^11^C]PK11195 PET images in the rat spinal cord injury model. (**C**) *(R)*-[^11^C]PK11195 uptake at the lesion level peaked at 4 days after SCI and was significantly higher compared to uptake in the control and sham groups. * *p* < 0.05.

**Figure 4 jcm-12-00116-f004:**
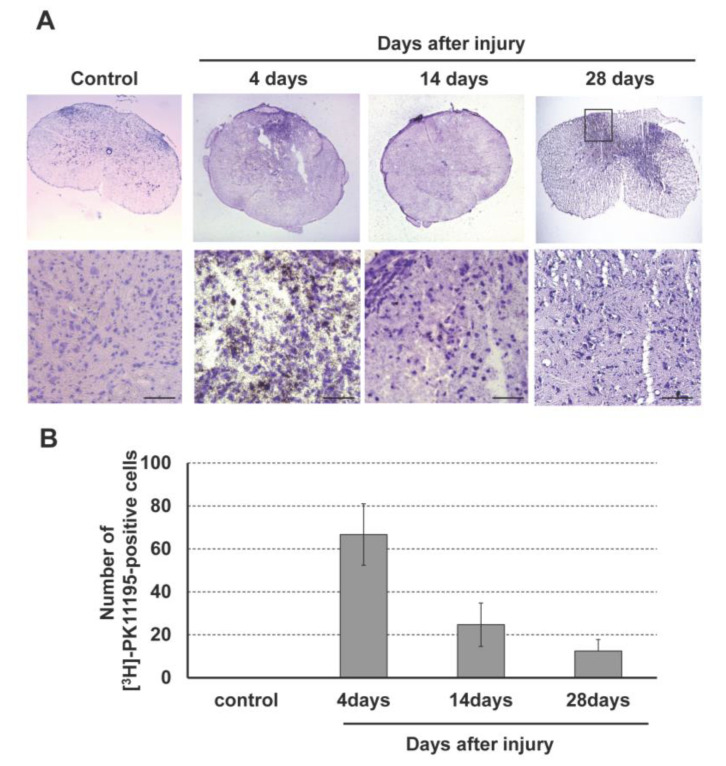
(**A**) *(R)*-[^3^H]PK11195 autoradiography in the rat spinal cord injury model. (**B**) Accumulation of *(R)*-[^3^H]PK11195 peaked at 4 days after SCI and then gradually decreased in the dorsal horn of the injured spinal cord. Scale bars = 100 μm.

**Figure 5 jcm-12-00116-f005:**
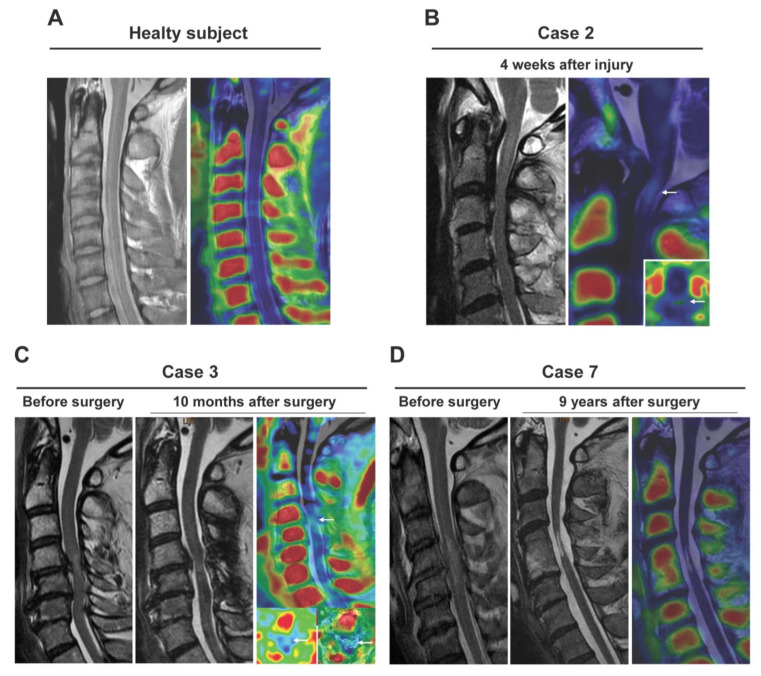
Clinical application of *(R)*-[^11^C]PK11195 PET/3T-MRI for patients with neuropathic pain. Data are shown for a healthy subject (**A**) and in patients examined at 4 weeks (**B**), at 10 months (**C**), and 9 years (**D**) after surgery. *(R)*-[^11^C]PK11195 uptake was apparent at 4 weeks and 10 months, but there was no significant *(R)*-[^11^C]PK11195 uptake at 9 years, despite the patient having severe neuropathic pain.

**Figure 6 jcm-12-00116-f006:**
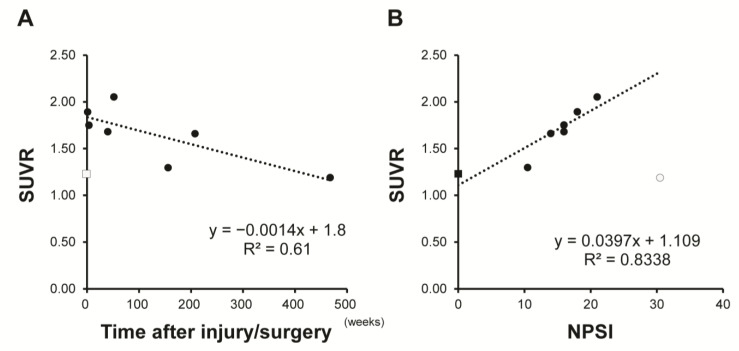
Relationship between standardized uptake value ratio (SUVR) and time after injury/surgery (**A**) and Neuropathic Pain System Inventory (NPSI) (**B**). SUVR was negatively correlated with time after injury/surgery and positively correlated with NPSI total score. Square markers: patients; Round marker: healthy subject; Outlined marker: excluded data for Pearson correlation analysis.

**Table 1 jcm-12-00116-t001:** Number of male Sprague Dawley rats used in each study.

	Time After Injury						Sham	Normal
	0 Day	1 Day	4 Days	7 Days	14 Days	28 Days		
Immunohistochemistry		3	7	3	3	3		
*(R)*-[^3^H]PK11195 autoradiography	3		3		3	3		
*(R)*-[^11^C]PK11195 PET imaging			5		3		3	3

**Table 2 jcm-12-00116-t002:** Summary of data for *(R)*-[^11^C]PK11195 FDG-PET/3T-MRI.

Case	Age	Sex	Disease	Time from Injury/Operation	NPSI	SUVR
Healthy subject	29	Male	-	-	0	1.23
1	71	Male	SCI	9 days	18	1.89
2	70	Male	OPLL + SCI	4 weeks	16	1.75
3	72	Male	OPLL	10 months	16	1.68
4	72	Male	CSM	1 year	21	2.05
5	66	Male	CSM	3 years	10.5	1.30
6	83	Female	OPLL + SCI	4 years	14	1.66
7	68	Male	SCI	9 years	30.5	1.19

NPSI: Neuropathic Pain Symptom Inventory; SUVR: standardized uptake value ratio; OPLL: ossification of the posterior longitudinal ligament; SCI: spinal cord injury; CSM: cervical spondylotic myelopathy.

## Data Availability

The data presented in this study are available on request from the corresponding author and subject to the ethical approvals in place and materials transfer agreements.

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
