# Peer review of "Evaluation of (R)-[11C]PK11195 PET/MRI for Spinal Cord-Related Neuropathic Pain in Patients with Cervical Spinal Disorders"

_jcm, 2022, doi:10.3390/jcm12010116_

Round 1
Reviewer 1 Report
The authors conducted an original study by assessing the potential of 11C-PK11195 PET in cervical spinal disorders, a field that remains poorly explored. I appreciated the clear and straightforward contribution combining an animal model and pilot study in patients.
However I’d like to point important lack of information, before the paper can be accepted for publication.
Throughout the text, the authors should clearly state the radiotracer they used. “PK11195” should not be used (as in the abstract). “[11C]PK11195” is not precise enough as racemic or enantiomere compounds can be radiosynthetized. The correct nomenclature for enantiomer (R) is (R)-[11C]PK11195. Please check the whole manuscript.
In the abstract: “No visualized uptake was detected in…” could be rephrased as “No uptake could be visualized in…”
In the introduction, it would be appreciable for the naive reader to get explanations about at-level and low-level classification. How does this refer to the population enrolled in the study?
The authors mentioned the study of Jeon et al (ref 21) on CRPS but there are a few other related studies that might also be of interest in the field of neuropathic pain: one paper on chronic fatigue with (R)-[11C]PK11195; 4 papers on chronic low back pain with [11C]PBR28… See Chauveau et al. Eur J Nucl Med Mol Imaging. 2021;49(1):201-220. And associated online database: https://www.zotero.org/groups/2578974/living_systematic_review_on_tspo_pet/library
Methods should be described with more details, especially:
-report weight of animals as range [min-max] rather than only the mean
-2% isoflurane does not provide a deep anesthenia, and more importantly, has no analgesic effect. The authors are strongly advised to modify their protocol for future studies (e.g. buprenorphine 0.05mg/kg)
-I would appreciate to have a better view of which animals were used for what. From the text, there are 27 SD rats used in total with 19 (at least) for immunohistochemistry, 12 for autoradiography, 14 for PET… I suggest clarifying the fate of each animal (in vivo and post mortem use) through a table
-Was CT performed during the same imaging session as PET in animals? Was the registration manual? Please report if any calibration procedure is done for animal PET (as for humans). Mention the actual estimated resolution for animal and human PET (not only pixel size), and add the volume of the ROIs – I understand they were drawn on a single slice, correct?
-The lack of normalization in animal data is intriguing: why not apply the same strategy as in humans, using a normal cervical spinal cord at an unaffected level?
-Please a paragraph for statistics: report results as mean/SD, and add 95 or 99% CI. State if any a priori sample size calculation was performed (or lack of, in the case of exploratory study). Describe statistical tests performed.
In the results, I suggest for better clarity to explicitly specify which immunostaining refers to which cell type (NeuN for neurons, etc). Figures 1-2-3-4 should mention the number of animals in each group (in the captions), and show individual points in addition to mean/SD. Captions from figures 3 and 4 should describe what panels A/B stand for.
Importantly, from figure 3A it is really difficult to figure out how ROI were drawn. If CT was used, then overlay PET/CT images should be shown as well. Is the increase on days 4 and 14 still detected after normalization?
On figure 5, to convince the reader, it would be useful to add a second case of positive detection in addition to case 3. I suggest reorganizing the figure so as to add a fourth panel.
If I understand correctly, figure 6 include the healthy subject on panel B but not on panel A, and identify case 7 as an outlier on panel B but not on panel A. Please be consistent between panels by using distinct symbols for healthy subject and case 7 vs the others. (Healthy subject could also be added on panel A using an arbitrary null absciss). Modify caption accordingly.
Discussion:
The animal studies cited (lines 274 sqq, ref 39-40) were also performed in the rat. So how the authors explain that “In the current study, background noise was less prominent in the rat model” (lines 280-1)? The vertebral signals are first described as potentially specific (hematopoietic stem cells in the medulla, line 280) and then qualified as “background noise” (line 301). the authors should clarify their thinking about this because in the latter case, using an alternative TSPO tracer might be beneficial for the detection of chronic inflammation in the spinal cord, but not in the former case?
Dec 13th,
Fabien Chauveau
Reviewer 2 Report
The manuscript by Kitade et al describes the evaluation of TSPO expression in a rat model of spinal cord injury and patients suffering from neuropathic pain related to cervical disorders by using [11C]PK11195 PET imaging. The manuscript is well written and findings are described clearly. Some minor comments authors may want to take into account are listed below.
1. For PET imaging, was the [11C]PK11195 used indeed a racemic mixture? Often, the R enantiomer is used, if that is the case here please specify throughout the manuscript.
2. For [11C]PK11195 used in animal experiments and patient studies, please provide synthetic details, e.g. molar activity and radiochemical purity.
3. On page 3, line 116, please also provide molar activity of [3H]PK11195 in GBq/umol.
4. More of an editing issue, but please check all headings carefully – whenever they start with [11C], something weird is happening to paragraph numbering (e.g. page 3 line 128).
5. In autoradiography experiments with [3H]PK11195, colocalization of [3H]PK11195 binding with immunofluorescent markers for microglia/astrocytes/neurons/oligodendrocytes or even GFP hematogenous cells would be helpful.
6. In figure 5, panel C, there is a typo in “9 years”.
7. Given the results of PET imaging in both rats and patients, and the statement made by the authors on page 11, lines 303-304, does PK11195 imaging mostly just show response to injury rather than a process involved in NeP?
8. Have authors considered looking at dynamics of microglial activation with respect to pro- or anti-inflammatory phenotype? Could be interesting to see over time if one phenotype dominates over the other during acute injury vs long lasting NeP?
